# Online Recognition of Fallen-Off Bond Wires in IGBT Modules

**DOI:** 10.3390/mi15030404

**Published:** 2024-03-17

**Authors:** Zhen Hu, Man Cui, Tao Shi

**Affiliations:** 1College of Automation, Nanjing University of Posts and Telecommunications, Nanjing 210023, China; huzhen0111@njupt.edu.cn; 2School of Information and Electronics, Beijing Institute of Technology, Beijing 100081, China; 7520210140@bit.edu.cn; 3Institute of Advanced Technology, Nanjing University of Posts and Telecommunications, Nanjing 210023, China

**Keywords:** bond wire, IGBT module, thermal management, reliability

## Abstract

As a core component of power conversion systems, insulated gate bipolar transistor (IGBT) modules continually suffer from severe thermal damage caused by temperature swings and shear stress, resulting in fatigue failure. Bond wires falling off is one of the failure modes of IGBT modules. Given that the number of fallen-off bond wires is a significant parameter to evaluate the health status of the IGBT modules, this paper proposes an online identification model to recognize the number of fallen-off bond wires during normal operation. Firstly, a database containing datum $V_{ce,on}-T_{j}-I_{C}$ (collector–emitter on-state voltage $V_{ce,on}$, chip junction temperature $T_{j}$, collector current $I_{C}$) planes with different fallen-off bond wires is built based on an offline aging test. Secondly, a Foster network model and a special circuit are designed to measure the junction temperature $T_{j}$ and the collector–emitter on-state voltage $V_{ce,on}$, respectively. Thirdly, the feature points of the IGBT module represented by $V_{ce,on}$, $T_{j}$, and $I_{C}$ are given to the database to recognize the number of fallen-off bond wires according to the position of the feature points in the datum plane. The experimental results show that the proposed method can determine the fallen-off bond wires under the operation condition.

## 1. Introduction

With the technological development of power semiconductor devices, power electronic systems have played an increasingly important role in efficient power conversion systems such as DC transmission, power supply, motor drive, microgrid, renewable energy generation, and energy storage. The reliability requirements of power electronic systems are getting higher [1,2]. Research reports on the reliability of power electronic systems show that the failure rate of power devices is the highest in converter systems, accounting for about 34% [3]. Therefore, the research on the operational reliability of power devices is a significant point in the reliability research of power electronic systems [4]. The insulated gate bipolar transistor (IGBT) module has been widely applied in power electronic systems due to its advantages in high operation frequency, fast switching speed, high input impedance, high current density, and low saturation voltage, leading electrical equipment into a new era of electric power electronics [5,6]. With the continuous improvement of current density and withstand voltage level of IGBT modules, the electric–thermomechanical load of IGBT modules is becoming heavier. The bond wire inside the module is the most vulnerable electrical connection part, and its reliability has attracted much attention [7,8].

The material thermal expansion coefficient at the contact point between the bond wires and the chip is diverse. Under the cyclic impact of temperature, the cracks form and extend at the bonding point, leading to the bond wire warpage or even falling off [9,10,11]. The external expressions are an increase in bonding resistance and a decrease in gate capacitance. Bond wires falling is the final degradation form of the IGBT module. Once a bond wire falls, other bond wires will bear more shear stress, causing the degradation process to accelerate [12,13,14]. The variations in bond resistance and gate capacitance lead to an increase in emitter saturation voltage, on-state loss, and junction temperature [13,14,15,16], affecting the thermal safety of the module. The bonding resistance of the IGBT module is small, on the order of micro-ohms, which is estimated according to the Miller platform height and collector current [15,16,17,18]. The gate capacitance of IGBT changes weakly with the bond wires falling off. There are only minor changes in gate capacitance even if all the bond wires fall off. Moreover, the monitoring of gate capacitance requires special measuring instruments, which is not easy to achieve under working conditions [19,20,21]. It is of great significance for the thermal management and safe operation of the power conversion system to study the online monitoring technology of bond wires under working conditions [22].

Many researchers have extensively studied the mechanism, evolution, and online monitoring technology of bond wire damage. A multifield simulation of electric–thermoforce coupling proceeded on the bond wires, pointing out that shear stress is the cause of fatigue damage to the bond wires. Only the effect of shear stress on the bond wire damage was studied, while not considering the influence of bond wires falling on the evolution of bond wire damage [23,24,25]. A monitoring method for bond wire damage based on the collector–emitter saturation voltage is proposed [26,27]. The voltage with specific collector current and zero temperature sensitivity characterize bond wire damage to avoid the influence of temperature on saturation voltage [28,29,30]. While the collector current is time-varying under variable operation conditions, leading to the monitoring conditions may not exist. A four-point monitoring method is proposed to measure bond resistance, while it can only be applied to open IGBT modules. The gate–emitter parasitic capacitance is used to determine the bond wire falling off [31,32,33], while the gate parasitic capacitance has a slight change with the fallen-off bond wires, and it is difficult to measure in practice. In addition, gate voltage $V_{G}$, gate current $I_{G}$, on-state resistance, etc., can also monitor bond wire damage [34,35,36]. Current research results have made considerable progress in monitoring bond wire damage and revealing the cause of bond wire damage, identifying the mode of bond wire damage.

Unfortunately, the present research results cannot achieve the accurate identification of the number of fallen-off bond wires. Given that the number of fallen-off bond wires is a significant parameter to evaluate the health status of the IGBT module, this paper proposes an online identification model to determine the number of fallen-off bond wires during the normal operation of the IGBT module. This work includes three aspects: (a) a database containing datum $V_{ce,on}-T_{j}-I_{C}$ (collector–emitter on-state voltage $V_{ce,on}$, junction temperature $T_{j}$, collector current $I_{C}$) planes with different fallen-off bond wires built based on an offline aging test; (b) a Foster network model and a special circuit designed to measure the junction temperature $T_{j}$ and the on-state voltage $V_{ce,on}$, respectively; and (c) the feature points of the IGBT module represented by $V_{ce,on}$, $V_{ce,on}$, and $I_{C}$, given to the database, as well as the number of fallen-off bond wires identified by the position of the feature points in the datum plane.

The remainder of this article is as follows. In Section 2, the online identification algorithm of the fallen-off bond wires is introduced. In Section 3, a database containing datum planes with different fallen-off bond wires is built. In Section 4, real-time measurements of junction temperature $T_{j}$ from the Foster model and on-state voltage $V_{ce,on}$ through a special circuit are introduced. In Section 5, the experimental analysis is employed to validate the effectiveness of the proposed method.

## 2. On-Line Identification of the Fallen-Off Bond Wires

The package of the welded IGBT module is in Figure 1. Direct Bonded Copper (DBC) consists of an upper copper layer, a ceramic plate, and a lower copper layer. On the one hand, it realizes the fixed and electrical connection between the IGBT chip and the continuous current diode. On the other hand, it forms the main channel for the module to dissipate heat. The connection of the DBC to the chip and the copper baseplate is solder. The connection between the chips and the external terminals is bond wires, and the module is filled with silicone gel to reduce the impact of external moisture, dust, and contamination.

During the operation, the power loss of the IGBT power module consists of switching loss and conduction losses, which are dissipated in the form of heat, resulting in temperature fluctuations inside the module. The coefficient of thermal expansion (CTE) of different materials in the structural layer is different. Thus, the thermal stress generates and makes the material fatigued. Eventually, the fatigue material leads to the failure of the IGBT module. The failure modes of welded IGBT modules are bond wire failure and solder failure.

Generally, aluminum-bond wires realize the electrical connection between the chip and the external terminals. The CTE between the aluminum and the silicon of the chip is different. The chip’s power loss and the bond wires’ joule heat increase the temperature inside the module, resulting in a temperature gradient and a shear stress between the contact point and the bond wires. In the long switching process, the fatigue deformation caused by shear stress accumulates, resulting in cracks in the contact points. The crack increases the thermal resistance, increasing joule heat. The increase in temperature gradient leads to more serious thermal damage to the bond wires. The above process is a positive feedback loop, as shown in Figure 2. The result is that the bond wires fall off.

The number of fallen-off bond wires is a significant parameter to evaluate the health condition of the IGBT module. Due to the enclosed package of the module, it is difficult to recognize the number of falloff bond wires based on conventional methods. The collector–emitter on-state voltage ($V_{ce,on}$) is a commonly used parameter to characterize the degradation of bond wires. Unfortunately, $V_{ce,on}$ cannot directly identify how many bond wires have fallen off. In addition, as temperature-sensitive electrical parameters (TSEPs), $V_{ce,on}$ is used to predict the chip junction temperature. Besides, $V_{ce,on}$ is affected by the collector current $I_{C}$. Therefore, the database consisting of $V_{ce,on}$, $T_{j}$, and $I_{C}$ is usually established, as shown in Figure 3. The chip junction temperature $T_{j}$ can be obtained by giving $V_{ce,on}$ and $I_{C}$ into the database.

The $V_{ce,on}-T_{j}-I_{C}$ plane in Figure 3 will have an offset when one bond wire falls off. Figure 4 shows the $V_{ce,on}-T_{j}-I_{C}$ planes of a healthy IGBT module and an aging IGBT module. The information of $V_{ce,on}$, $T_{j}$, and $I_{C}$ represents the operation characteristics of the IGBT module. The values of $V_{ce,on}$, $T_{j}$, $I_{C}$ determine where the feature point $F_{\left(V_{ce,on}-T_{j}-I_{C}\right)}$ is located in the database of Figure 4. The $V_{ce,on}-T_{j}-I_{C}$ plane where $F_{\left(V_{ce,on}-T_{j}-I_{C}\right)}$ is located decides how many bond wires fall off in the IGBT module. There are two significant steps to confirm the location of $F_{\left(V_{ce,on}-T_{j}-I_{C}\right)}$: one is to obtain the $V_{ce,on}-T_{j}-I_{C}$ plane of different fallen-off bond wires; the other is to obtain the $V_{ce,on}$, $T_{j}$, and $I_{C}$ information in real time during the normal operation of the IGBT module. With the combination of the two steps, the number of fallen-off bond wires can be identified. Next, we introduce how to create a $V_{ce,on}-T_{j}-I_{C}$ plane with different fallen-off bond wires and how to obtain $V_{ce,on}$, $T_{j}$, and $I_{C}$ information in real time.

## 3. Datum Planes with Different Fallen-Off Bond Wires

An experimental test system is applied to obtain the $V_{ce,on}-T_{j}-I_{C}$ datum planes with different fallen-off bond wires. The experimental equipment is in Figure 5, including a commercial IGBT module produced by SEMIKRON (SKM75GB12T4), an oscilloscope for measuring various electrical signals of the module, a low-power DC supply for offering a driving signal, a high-power programmable DC supply for the test current, and a temperature chamber to keep the module at a constant temperature.

The experimental process is as follows:
(1)The DC voltage source of 15 V is applied to the IGBT gate terminal ($V_{ge}$) during the test. Different gate voltages affect the value of $V_{ce,on}$. The effect of gate voltage should be considered when establishing the reference plane. Considering that for a specific application such as wind power generation, the gate voltage from the driver is constant. We only need to establish the reference plane according to the gate voltage from the driver. In this experiment, we take a 15 V gate voltage as an example to illustrate the establishment process of the reference plane.(2)The IGBT module is placed in the temperature chamber for 3–5 min to keep the junction temperature of the module consistent with the temperature chamber in a stable status. The upper package of the IGBT module located in the temperature chamber is removed. The junction temperature is measured by an infrared camera to determine whether it reaches the same temperature as the temperature chamber. If not, the IGBT module remains in the temperature chamber until reaching the set temperature. The temperature of the chamber varies from 35 °C to 110 °C in 15 °C intervals, i.e., the junction temperature of the IGBT module varies from 35 °C to 110 °C in 15 °C intervals.(3)The high-power programmable DC supply is set as a pulsed-current generator to provide high-current pulses to the IGBT module. In the pulsed-current test, a series of five high-current pulses with different magnitudes are applied to the collector terminal of the module at various temperatures of 35 °C, 50 °C, 65 °C, 80 °C, 95 °C, and 110 °C. Each of the pulses had a duration of 0.5 s. The magnitude of the pulsed current varies from 30 A to 70 A in 10 A intervals. The pulses were 0.5 s apart. The oscilloscope in the test was Keysight’s InfiniiVision 2000-X series, with sampling rates up to 2 GSa/s and memory of 1 Mpt per channel. During the test, the sampling time is 5 s, the oscilloscope real-time sampling rate is 0.2 MSa/s, and the amount of collected $V_{ce,on}$ is 20 K. We take the average of the first twenty $V_{ce,on}$ as the reference value. In this way, $V_{ce,on}$ is not affected by the heat caused by the current pulse.(4)The collector current $I_{C}$, the collector–emitter on-state voltage $V_{ce,on}$, and the junction temperature $T_{j}$ are recorded. Accordingly, the datum plane of $V_{ce,on}-T_{j}-I_{C}$ is received.(5)The enclosed package of the IGBT module is removed, and the bond wire is cut intentionally, repeating the above (1)–(3) process each time when cutting one bond wire. The above process is finished when three bond wires are broken. Then, the $V_{ce,on}-T_{j}-I_{C}$ datum planes with different fallen-off bond wires are obtained, as shown in Figure 6.

## 4. Real-Time Measurement of Junction Temperature and On-State Voltage

### 4.1. On-Line Measurement of Junction Temperature Based on Foster Network Model

The junction temperature extraction methods of IGBT modules consist of four categories, which are the physical contact method, optical method, thermal network method, and temperature-sensitive electrical parameters (TSEPs) method. The physical contact method receives the junction temperature by setting a thermocouple or thermistor around the chip. The optical method uses an infrared thermal imaging camera to measure the junction temperature. The above two methods need to invade the module or destroy the module package, which brings inaccuracy to the temperature measurement and cannot reflect the junction temperature of the IGBT module in actual working conditions.

The physical mechanism of the TSEPs method is that some physical parameters inside the device change with the junction temperature. For example, the intrinsic carrier concentration and carrier lifetime increase with the increase in temperature, and the carrier mobility decreases with the increase in temperature. The change in internal physical parameters with junction temperature causes the corresponding electrical parameters of the device to shift, and the external performance is the change in on-state resistance/voltage, on/off delay time, voltage/current change rate, and other parameters. These electrical parameters that change with the junction temperature are called TSEPs. Different TSEPs have advantages and disadvantages in sensitivity, linearity, robustness, etc. It is necessary to select appropriate TSEPs according to device type and actual operating conditions to extract junction temperature. However, most TSEPs are affected by bond wire degradation, and the accuracy of junction temperature measurement decreases with the degradation degree of bond wires. In this paper, the TSEPs method is not suitable for measuring chip junction temperature.

The thermal network model predicts the junction temperature only based on the information of the module’s power loss and the case temperature, which has the advantages of simple measurement and low hardware requirements. The thermal network model consists of the Foster model and the Cauer model. The parameters of the Foster model are received by fitting the transient thermal impedance curve with the least-square method, and each temperature node has no physical meaning. The parameters of the Cauer model are recognized using the physical property of the packaging material, and each temperature node represents the temperature of each layer. Compared with the Cauer model, the Foster model has the advantages of high precision in junction temperature prediction and low difficulty in parameter recognition. Therefore, the Foster model is more common in practical applications. The Foster model is adopted to predict junction temperature in this paper.

The chip is the heat source of the IGBT module. The heat is generally generated on the upper surface of the chip and transferred from the chip to the baseplate through various layers with different materials, generating the thermal path of the module, as shown in Figure 7. The transient thermal impedance from chip to baseplate $Z_{JC}\left(t\right)$ is commonly used to characterize the thermal path. The formula of $Z_{JC}\left(t\right)$ is as follows:(1)$$Z_{JC}\left(t\right)=\frac{T_{J}\left(t\right)-T_{C-chip}\left(t\right)}{P}$$
where *P* is the total power loss of the module, $T_{J}\left(t\right)$ is the chip junction temperature, and $T_{C-chip}\left(t\right)$ is the case temperature of the baseplate.

The function of $Z_{JC}\left(t\right)$ is described by an electrical equivalent resistance–capacitance (RC) network, shown in Figure 8, which is known as the Foster network. A series of exponential terms is used to characterize the time response of the Foster network as follows:(2)$$Z_{JC}\left(t\right)=\sum_{i=1}^{n}R_{i}\left(1-e^{-t/R_{i}C_{i}}\right).$$

As an equivalent circuit model, the parameters of the Foster network are fitted from the transient thermal impedance $Z_{JC}\left(t\right)$ based on the least-square method. The $Z_{JC}\left(t\right)$ curve is easily received by finite element analysis (FEA). A three-dimensional (3D) model based on the dimensions and materials of the module is modeled by a map-making software, i.e., Pro/Engineer (Version 5.1), as shown in Figure 9. The 3D IGBT model is given to a commercial FEA software (ANSYS, Version 17.2) to process a transient thermal analysis.

The total power losses of the IGBT module are composed of conduction loss and switching loss and are estimated as follows:(3)$$\left\{\begin{array}{cc} P & =P_{cond}+P_{sw}\\ P_{cond} & =V_{ce,on}\times I_{C}\\ P_{sw} & =\left(E_{on}+E_{off}\right)\times f_{sw} \end{array}\right.$$
where $P_{cond}$ is the conduction power loss, $P_{sw}$ is the switching power loss, $I_{C}$ is the collector current, $E_{on}$ and $E_{off}$ are the turn-on and turn-off energy of the module, respectively, and $f_{sw}$ is the switching frequency. The transient thermal analysis proceeded by placing the power losses on the IGBT chip. The results of the thermal analysis for the IGBT module are received, as shown in Figure 10.

By substituting the simulation results into (Equation 1), the transient thermal impedance curve $Z_{JC}\left(t\right)$ for the IGBT module is derived, as shown in Figure 11. A fourth-order Foster network has been proven to have a good approximation for the transient thermal impedance. The values of the RC component are received through the least-square method, as shown in Table 1. As a result, the Foster network model was built to predict the junction temperature of the IGBT model in real time.

### 4.2. On-Line Measurement of Collector–Emitter On-State Voltage

The collector–emitter voltage $V_{ce}$ varies from hundreds of volts to several volts per cycle, making it difficult for traditional circuits to measure on-state voltage $V_{ce,on}$. A special circuit for measuring on-state voltage is designed in this paper, and the schematic of the measurement circuit is in Figure 12.

By properly selecting the gain of the amplifier, as $R_{1}=R_{2}$, the output of the amplifier can be expressed as follows:(4)$$V_{op-amp}=V_{b}-\left(\left(V_{a}-V_{b}\right)\cdot \frac{R_{2}}{R_{1}}\right)=2V_{b}-V_{a}=V_{ce,on}$$

Thus, the $V_{ce,on}$ is acquired online through the special circuit.

## 5. Experimental Validation

In this section, the effectiveness of the improved thermal equivalent circuit model is validated by an experimental case. The test equipment is in Figure 13, including a commercial IGBT module produced by SEMIKRON (SKM75GB12T4, the upper package is removed), an IR camera to measure the chip junction temperature, a recorder for obtaining various electrical signals of the module, a signal generator for offering a driving signal, a DC power supply for the test current, and air-cooled equipment to cool the module, a National Instruments (NI) data acquisition instrument for measuring the case temperatures, a temperature chamber to keep the module at a constant temperature.

To obtain the evolution of various signals that characterize the IGBT’s degradation, we built an accelerated aging test platform for power devices based on the experimental equipment shown in Figure 13, applying a 10% overload current to the IGBT module to heat the junction temperature to 180 °C quickly and then bring down the junction temperature to 60 °C through the air-cooled equipment. In this way, the IGBT module is subject to a 120 °C temperature swing in one thermal cycle, resulting in thermal damage to bond wires. The test process needs to pay attention to three aspects: (1) The accelerated aging test of the IGBT module is stopped when one bond wire is observed to fall off, and the experimental process in Section 3 starts to obtain the $V_{ce,on}-T_{j}-I_{C}$ plane. (2) $V_{ce,on}$ is measured simultaneously through the special circuit in Section 4.2 and the recorder. (3) The IGBT accelerated aging test is terminated when two bond wires are observed to fall off.

The performance of the Foster network model in predicting junction temperature needs to be validated before the accelerated aging test. A DC pulse current is applied to the IGBT module to raise the chip junction temperature. The module’s power loss is estimated according to the electrical signal collected by the recorder. The power loss is given to the Foster network model to predict the junction temperature. The temperature results from the model were compared with those measured by the IR camera, as shown in Figure 14. The IR camera used in this paper is the FOTRIC series, which can continuously collect temperature at a point and obtain a temperature map. Figure 15 shows the temperature distribution of the module’s upper surface. From Figure 15, the temperature at the chip is the highest. The farther away from the chip, the cooler it gets. That is because the chip is the heat source of the IGBT module. From Figure 14, the temperature results from the Foster network model can accurately track the temperature results from the IR camera. The difference between the two temperature results was generally less than 3 °C. The correlation between the two signals is more than 0.95, indicating that the Foster network model can predict chip junction temperature accurately. The RC parameters of the Foster model are fitted from the transient thermal impedance curve based on the least-squares method. There are some errors between the RC parameters obtained by the fitting means and the real RC parameters of the IGBT module. Therefore, the junction temperature predicted by the Foster model is slightly less than that of the IR camera.

The measured $V_{ce,on}$ from the designed circuit and the recorder is in Figure 16. The starting point of the thermal cycle in Figure 16 is 0, meaning that the IGBT module is a healthy device without thermal damage. Thermal damage occurs to the bond wires when the IGBT module goes through a thermal cycle. The IGBT fails when the thermal damage of bond wires accumulates to the threshold value. From Figure 16, the thermal damage to bond wires obtains very slow growth before the 3000 thermal cycles. Therefore, the values of $V_{ce,on}$ change slowly. According to the theory of cumulative damage in fatigue, the thermal damage of bond wires reaches the threshold after 3000 cycles. Accordingly, $V_{ce,on}$ has rapid growth after 3000 cycles. When one bond wire falls off, $V_{ce,on}$ increases exponentially. In addition, the $V_{ce,on}$ results from the designed circuit can accurately track the results from the recorder. The difference between the two results was generally less than 2%, indicating that the designed circuit can accurately measure $V_{ce,on}$.

The accelerated aging test of the IGBT module is stopped when one bond wire is observed to fall off, and the experimental process in Section 3 starts to obtain the $V_{ce,on}-T_{j}-I_{C}$ plane. This $V_{ce,on}-T_{j}-I_{C}$ plane is compared with the datum planes with different fallen-off bond wires in Section 3, as shown in Figure 17. From Figure 17, taking the first fallen-off bond wire as an example, there is a little difference between the $V_{ce,on}-T_{j}-I_{C}$ plane from the accelerated aging test and the datum plane in the database. This is because the accelerated aging test had been running for a while when we observed the bond wire breaking, resulting in a further degradation of the bond wire. The number of fallen-off bond wires can be recognized by comparing the positions of the two planes. In practice, the whole $V_{ce,on}-T_{j}-I_{C}$ plane is not necessary; the identification of the fallen-off bond wires can be completed through the relative position of the feature point ($F_{\left(V_{ce,on}-T_{j}-I_{C}\right)}$) and the datum plane.

## 6. Discussion

$V_{ce,on}$ is considered as a failure precursor for the bond wire failure; the application of $V_{ce,on}$ to real-time field monitoring remains challenging. V. Smet et al. proposed a unit in an intelligent power module (IPM) to periodically monitor the device’s $V_{ce,on}$ and current $I_{C}$, which are sampled simultaneously [37]. For a given current value, the measured $V_{ce,on}$ is compared against a value saved in a look-up table. The deviation is used as an indication of the integrity of the bond wires and the level of deterioration due to power cycling. Unfortunately, this method can only obtain the device’s overall fatigue status and cannot observe the aging details of the bond wires. Another issue associated with $V_{ce,on}$-based condition-monitoring techniques is that $V_{ce,on}$ is affected by both junction temperature and health conditions of IGBT modules [38]. Considering the above-mentioned problems, this paper proposes a model to investigate the damage details of the bond wires inside the device, i.e., the number of liftoff bond wires. A database containing datum $V_{ce,on}-T_{j}-I_{C}$ planes with different liftoff bond wires is built based on an offline aging test. The device’s feature points represented by $V_{ce,on}$, $T_{j}$, and $I_{C}$ are given to the database to recognize the number of liftoff bond wires according to the position of the feature points in the datum plane. Meanwhile, a Foster network model is designed to predict the junction temperature $T_{j}$ in real time while measuring $V_{ce,on}$. Accordingly, the information of $V_{ce,on}$ and $T_{j}$ is consistent. The influence of $T_{j}$ on $V_{ce,on}$ can be removed. This research is of great help in exploring the internal fatigue condition of IGBT modules and realizing more detailed maintenance management of power converters.

## 7. Conclusions

In this paper, an online identification model is proposed to determine the number of fallen-off bond wires during the normal operation of the IGBT module. This work includes three aspects of work. A database containing datum $V_{ce,on}-T_{j}-I_{C}$ planes with different fallen-off bond wires is built based on an offline aging test. A Foster network model is proposed to predict the chip junction temperature of the IGBT model, and a special circuit is designed to measure the collector-emitter on-state voltage. The feature points consisting of $V_{ce,on}$, $V_{ce,on}$, and $I_{C}$ are given to the database to recognize the number of fallen-off bond wires through the position of the feature points in the datum plane. The experimental results validate the effectiveness of the proposed method. The Foster network model can accurately receive the chip junction temperature with an error of about 3 °C, and the designed circuit can accurately measure the value of $V_{ce,on}$ with an error of less than 2%. In addition, the number of fallen-off bond wires is accurately recognized through the position of the feature points $F_{\left(V_{ce,on}-T_{j}-I_{C}\right)}$ in the datum plane with different fallen-off bond wires.

## Figures and Tables

**Figure 1 micromachines-15-00404-f001:**
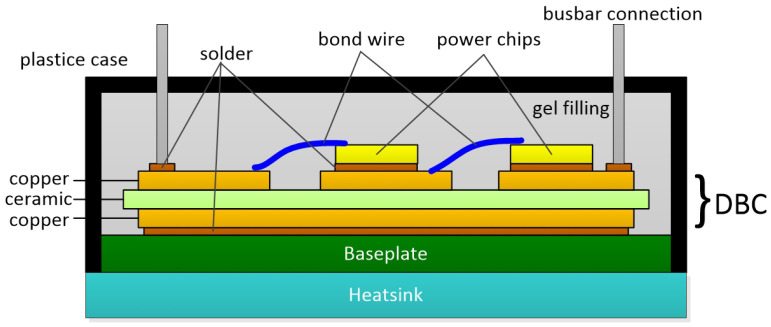
The package of a welded IGBT module.

**Figure 2 micromachines-15-00404-f002:**
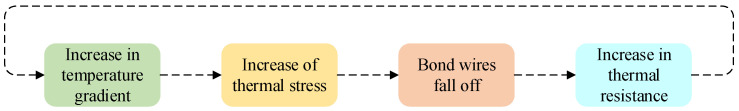
A positive feedback loop of bond wires degradation.

**Figure 3 micromachines-15-00404-f003:**
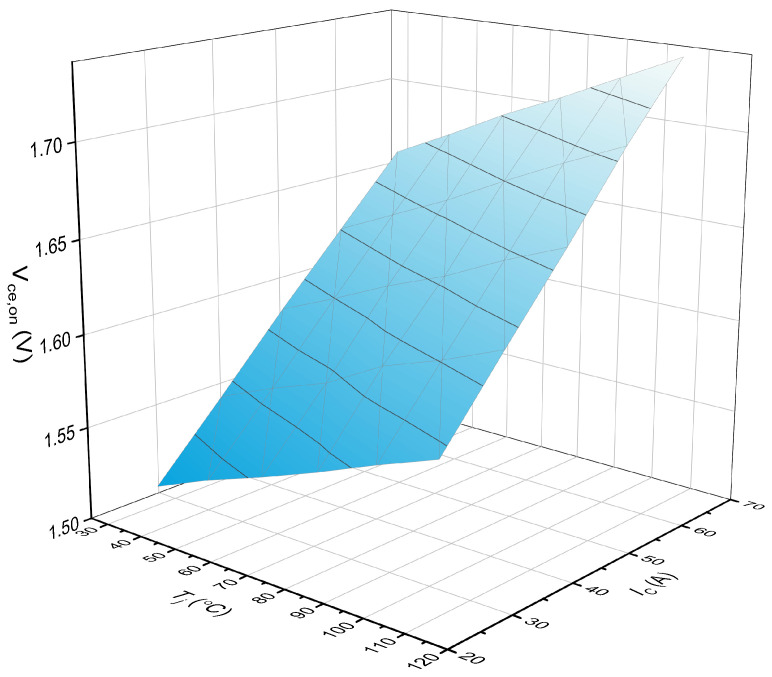
The $V_{ce,on}-T_{j}-I_{C}$ plane of the IGBT module.

**Figure 4 micromachines-15-00404-f004:**
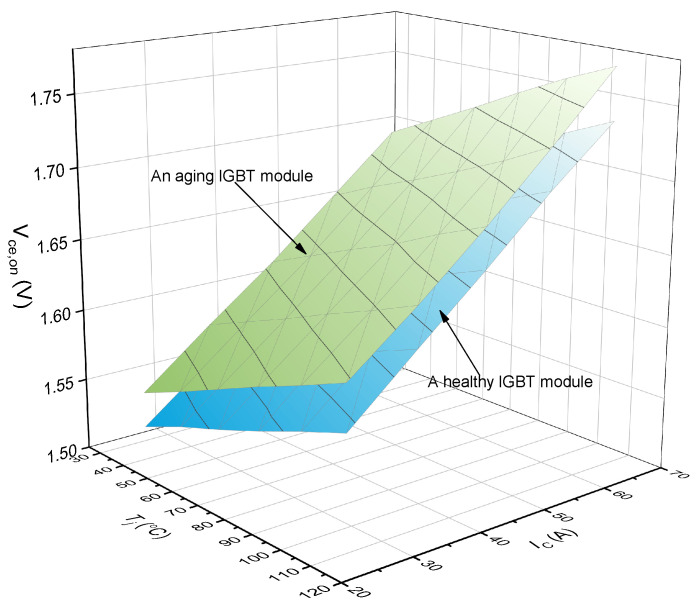
The $V_{ce,on}-T_{j}-I_{C}$ planes of a healthy IGBT module and an aging IGBT module.

**Figure 5 micromachines-15-00404-f005:**
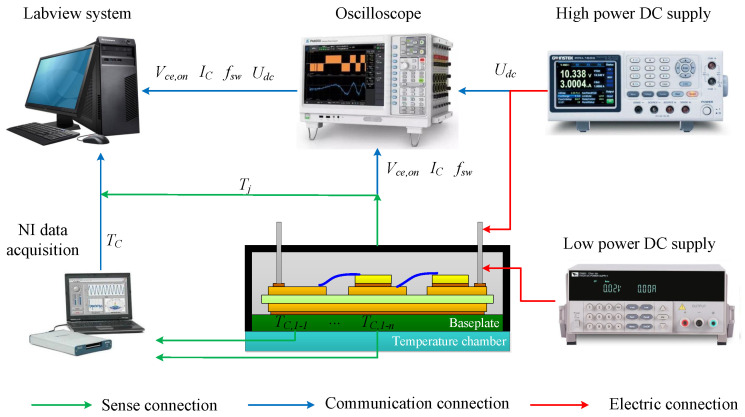
The experimental test system.

**Figure 6 micromachines-15-00404-f006:**
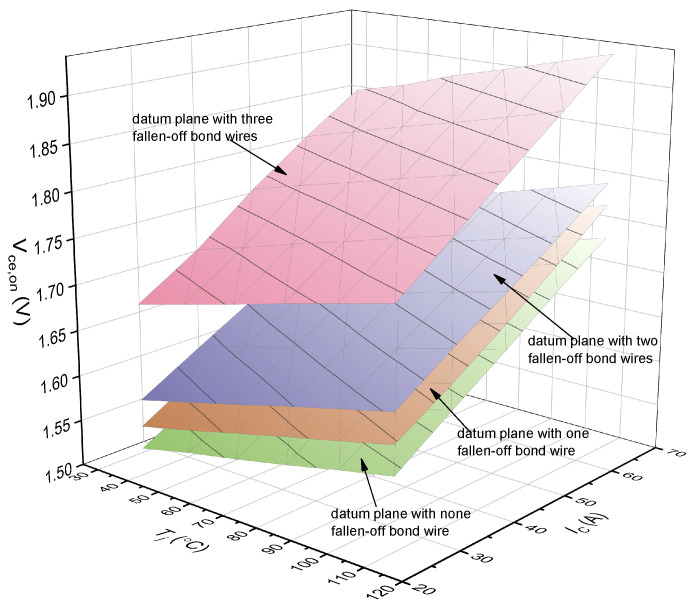
The $V_{ce,on}-T_{j}-I_{C}$ datum planes with different fallen-off bond wires.

**Figure 7 micromachines-15-00404-f007:**
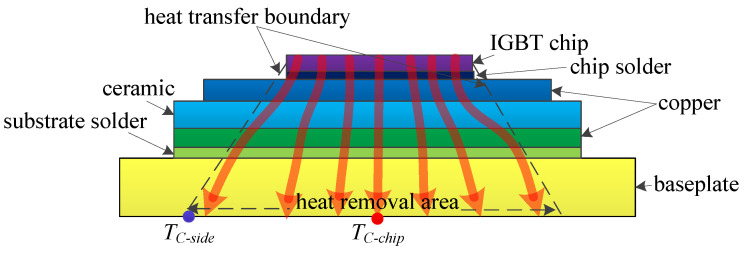
The thermal path of a healthy IGBT module.

**Figure 8 micromachines-15-00404-f008:**
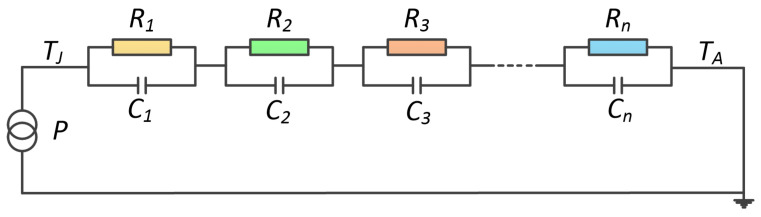
The Foster network model.

**Figure 9 micromachines-15-00404-f009:**
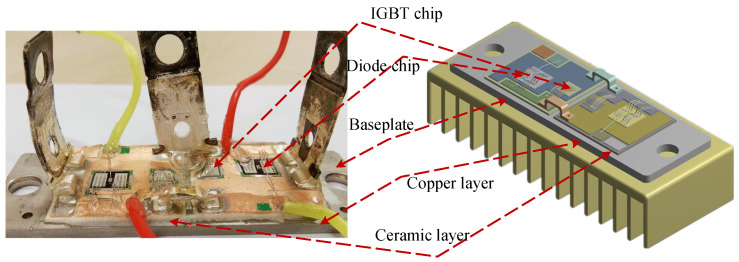
The three-dimensional (3D) model of the IGBT module.

**Figure 10 micromachines-15-00404-f010:**
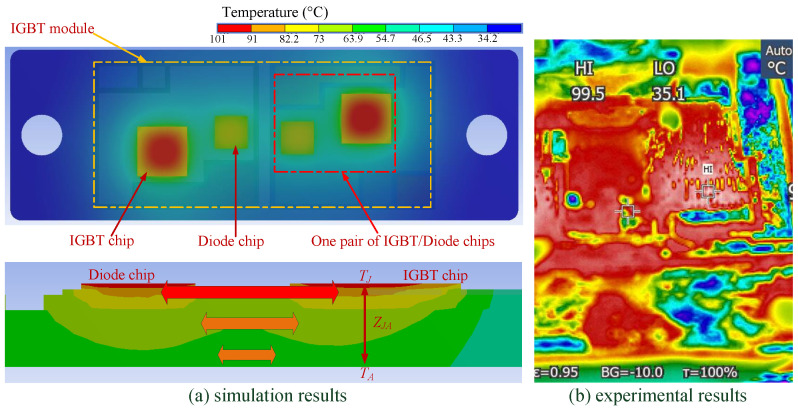
The thermal analysis results of the IGBT module.

**Figure 11 micromachines-15-00404-f011:**
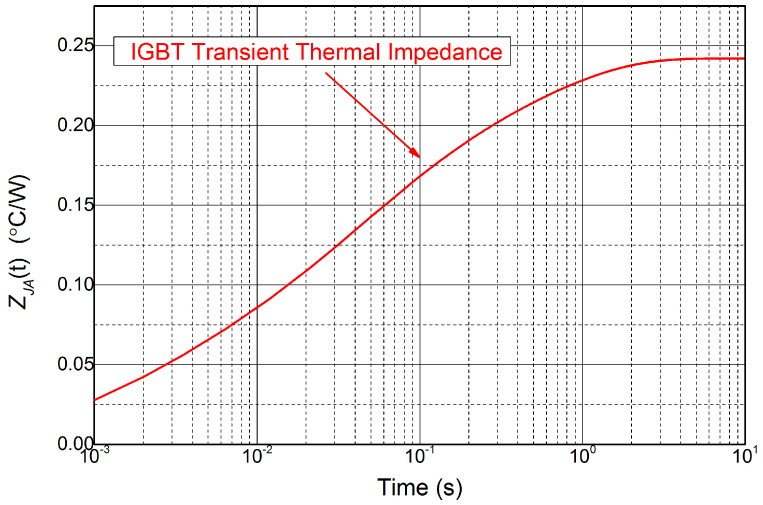
The transient thermal impedance curve of the IGBT module.

**Figure 12 micromachines-15-00404-f012:**
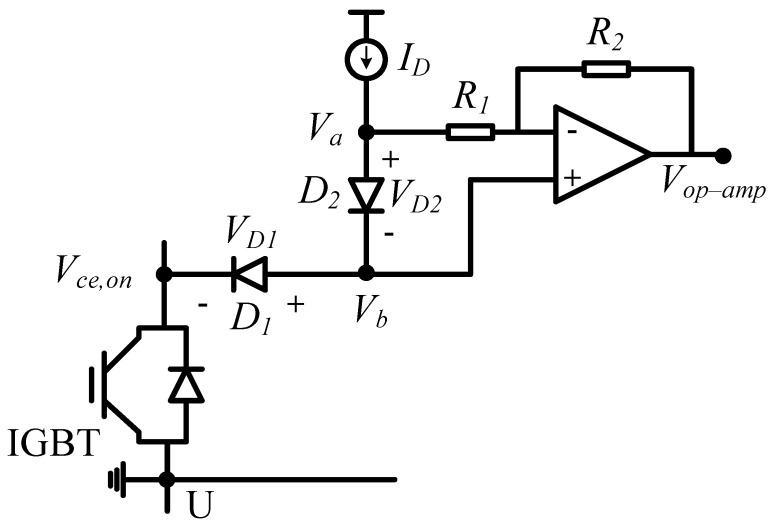
Schematic of the $V_{ce,on}$ measurement circuit.

**Figure 13 micromachines-15-00404-f013:**
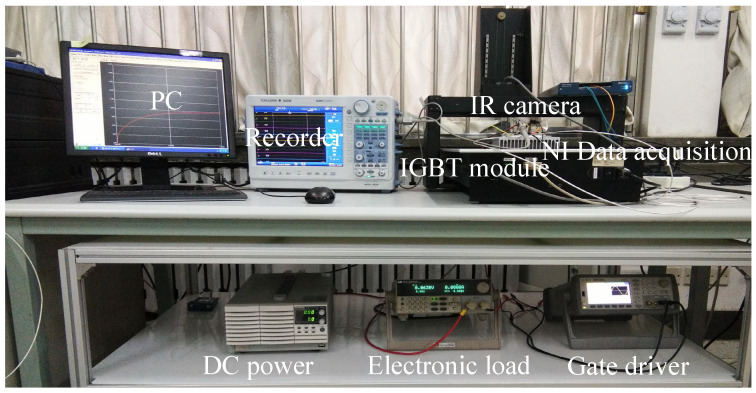
Experimental setup.

**Figure 14 micromachines-15-00404-f014:**
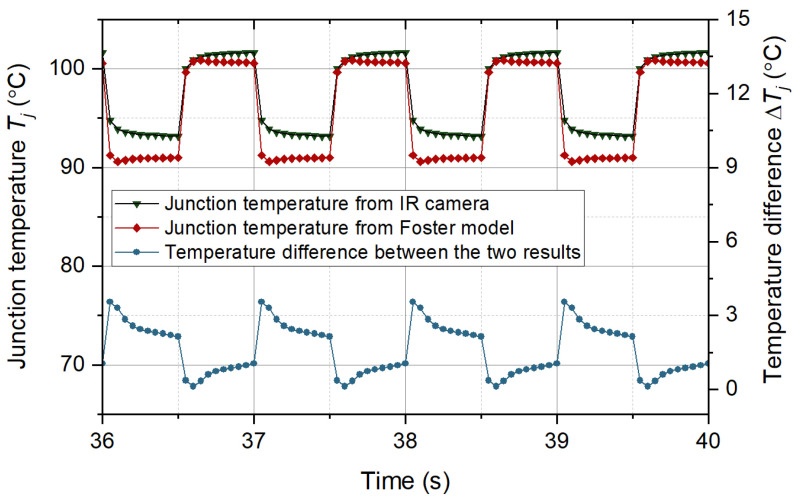
The junction temperatures from IR camera and the Foster network model.

**Figure 15 micromachines-15-00404-f015:**
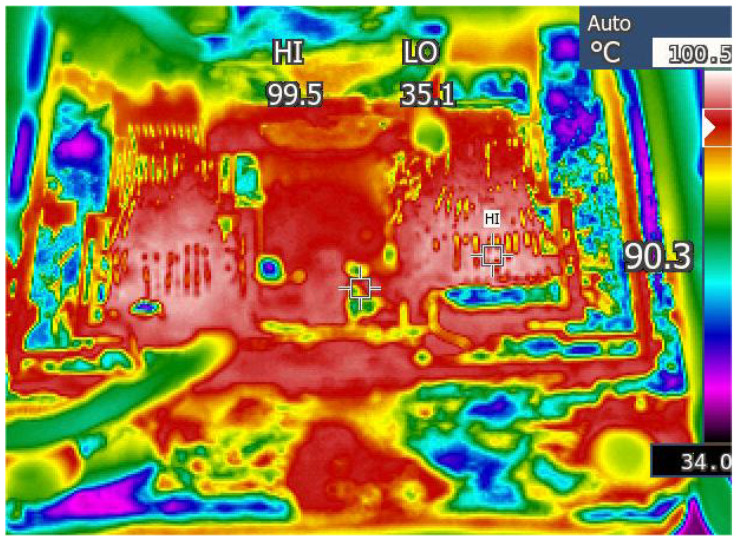
The temperature results from the IR camera.

**Figure 16 micromachines-15-00404-f016:**
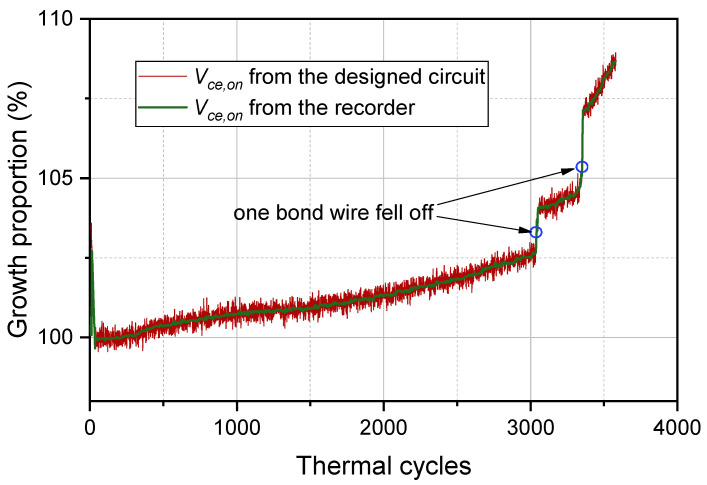
The measured $V_{ce,on}$ results from the designed circuit and the recorder.

**Figure 17 micromachines-15-00404-f017:**
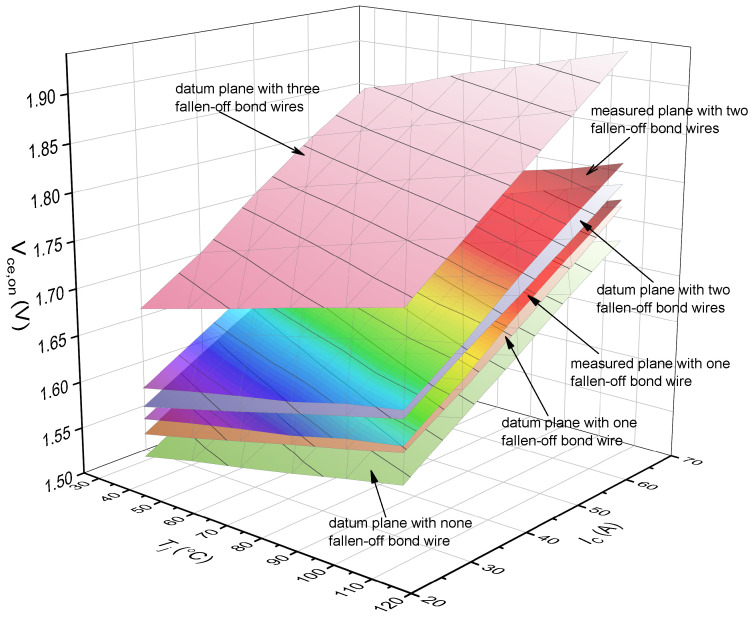
The experimental results of $V_{ce,on}-T_{j}-I_{C}$ planes.

**Table 1 micromachines-15-00404-t001:** The Foster network model parameters.

i	1	2	3	4
$R_{i}$ (K/W)	0.18	0.064	0.022	0.004
$C_{i}$ (J/K)	0.182	0.75	0.36	1.25

## Data Availability

Data is contained within the article.

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
