# Peer review of "Online Recognition of Fallen-Off Bond Wires in IGBT Modules"

_micromachines, 2024, doi:10.3390/mi15030404_

Round 1

Reviewer 1 Report

Comments and Suggestions for Authors

This manuscript describes the online recognition of the fallen -off bond wires in the IGBT modules. The devices in this work can play an important role in the power conversion system. The content is suitable for this journal and the length is enough. A minor revision is suggested for its publication in in this journal.

1.     The time variation of the aging process for the IGBT module should be pointed out, when comparing performances with those of the healthy module. Additionally, how about the effect of the different time on the performances? Please provide more evidences.

2.     The illustrations of the each part in the IGBT module should be provided in the Figure 9. Please revise the Figure 9.

3.     The junction temperatures from IR camera and the Foster network model are different, please clarify the reason.

4.     Some progress towards electronics are suggested to be included in the reference part, such as Nat Rev Mater 8, 587–603 (2023); https://doi.org/10.1007/s12598-022-02133-8; https://doi.org/10.1007/s12598-022-02166-z

Reviewer 2 Report

Comments and Suggestions for Authors

An online identification model is proposed to determine the number of fallen-off bond wires. Some critical questions should be solved.

1. When establishing the reference plane, the gate voltage is only set to 15V, but different gate voltages will change Vce,on, this condition is not complete.

2. Whether 3-5 minutes in the temperature box can ensure that the IGBT chip temperature is the same as the heating temperature, needs to be verified. 

3. When testing different currents, each pulse lasts for 0.5 s, and the IGBT chip will warm up, which is not consistent with the external given temperature, which time node is chosen to extract the stable temperature value, there is an impact on the accuracy of the reference plane.

4. When extracting Vce,on, is it extracted during the on-warming phase or the off phase of the IGBT module? The time point of extraction needs to be taken seriously because of the temperature dependence.

5. Based on the constructed reference plane for bond wire degradation, only the case of complete bond wire break-off can be evaluated, but actually, the Vce,on would reach the failure threshold in the case of only cracking or incomplete detachment of bond wires. The practical usability of the proposed method is controversial.

6. Please add a comparative analysis and discussion section to highlight the contribution of the article with a comprehensive comparison with existing methods.

Comments on the Quality of English Language

Moderate editing of the English language is required.

Round 2

Reviewer 2 Report

Comments and Suggestions for Authors

Thanks to the authors for their careful revisions and replies. My question has been answered satisfactorily. Another small suggestion is to add some references of IGBT application scenarios in the introduction section, and the related fault diagnosis and fault tolerance research also supports the importance of IGBT reliability. The following literature is listed for citation reference.

1. D. Xie, C. Lin, H. Lin, W. Liu, Y. Du and T. Basler, "OC Switch Fault Diagnosis, Pre- and Postfault DC Voltage Balancing Control for a CHBMC Using SVM Concept," in IEEE Transactions on Power Electronics, vol. 39, no. 1, pp. 677-692, Jan. 2024, doi: 10.1109/TPEL.2023.3319136.

2. A. Raki, Y. Neyshabouri, M. Aslanian and H. Iman-Eini, "A Fault-Tolerant Strategy for Safe Operation of Cascaded H-Bridge Multilevel Inverter Under Faulty Condition," in IEEE Transactions on Power Electronics, vol. 38, no. 6, pp. 7285-7295, June 2023, doi: 10.1109/TPEL.2023.3257278.

3. H. Lin, C. Lin, D. Xie, P. Acuna and W. Liu, "A Counter-Based Open-Circuit Switch Fault Diagnostic Method for a Single-Phase Cascaded H-Bridge Multilevel Converter," in IEEE Transactions on Power Electronics, vol. 39, no. 1, pp. 814-825, Jan. 2024, doi: 10.1109/TPEL.2023.3324871.

4. Ó. López et al., "Postfault Operation Strategy for Cascaded H-Bridge Inverters Driving a Multiphase Motor," in IEEE Transactions on Industrial Electronics, vol. 71, no. 5, pp. 4309-4319, May 2024, doi: 10.1109/TIE.2023.3281688.
